Fasciola gigantica, F. hepatica and Fasciola intermediate forms: geometric morphometrics and an artificial neural network to help morphological identification

Sumruayphol Suchada 1
Siribat Praphaiphat 2
Dujardin Jean-Pierre 3
Dujardin Sébastien 3
Komalamisra Chalit 4
Thaenkham Urusa urusa.tha@mahidol.ac.th 5
1 Department of Medical Entomology, Faculty of Tropical Medicine, Mahidol University , Bangkok , Thailand
2 Department of Biochemistry, Faculty of Science, Mahidol University , Bangkok , Thailand
3 IRD, UMR, INTERTRYP IRD, CIRAD, University of Montpellier , Montpellier , France
4 Mahidol-Bangkok School of Tropical Medicine, Faculty of Tropical Medicine, Mahidol University , Bangkok , Thailand
5 Department of Helminthology, Faculty of Tropical Medicine, Mahidol University , Bangkok , Thailand
Wilson Laura
Electronic publication date: 2020 Feb 18
Publication date: 2020
Volume: 8
Electronic Location ID: e8597
Received 2019 Aug 8; Accepted 2020 Jan 18
Copyright: ©2020 Sumruayphol et al.
Copyright year: 2020
Copyright holder: Sumruayphol et al.
License: This is an open access article distributed under the terms of the Creative Commons Attribution License, which permits unrestricted use, distribution, reproduction and adaptation in any medium and for any purpose provided that it is properly attributed. For attribution, the original author(s), title, publication source (PeerJ) and either DOI or URL of the article must be cited.
License URL: https://creativecommons.org/licenses/by/4.0/

Keywords: Fasciola, Molecular identification, ITS1&ITS2 markers, Morphometrics, Artificial neural networks

Funding: The authors received no funding for this work.

==============================
Background

Fasciola hepatica and F. gigantica cause fascioliasis in both humans and livestock. Some adult specimens of Fasciola sp. referred to as “intermediate forms” based on their genetic traits, are also frequently reported. Simple morphological criteria are unreliable for their specific identification. In previous studies, promising phenotypic identification scores were obtained using morphometrics based on linear measurements (distances, angles, curves) between anatomical features. Such an approach is commonly termed “traditional” morphometrics, as opposed to “modern” morphometrics, which is based on the coordinates of anatomical points.

Methods

Here, we explored the possible improvements that modern methods of morphometrics, including landmark-based and outline-based approaches, could bring to solving the problem of the non-molecular identification of these parasites. F. gigantica and Fasciola intermediate forms suitable for morphometric characterization were selected from Thai strains following their molecular identification. Specimens of F. hepatica were obtained from the Liverpool School of Tropical Medicine (UK). Using these three taxa, we tested the taxonomic signal embedded in traditional linear measurements versus the coordinates of anatomical points (landmark- and outline-based approaches). Various statistical techniques of validated reclassification were used, based on either the shortest Mahalanobis distance, the maximum likelihood, or the artificial neural network method.

Results

Our results revealed that both traditional and modern morphometric approaches can help in the morphological identification of Fasciola sp. We showed that the accuracy of the traditional approach could be improved by selecting a subset of characters among the most contributive ones. The influence of size on discrimination by shape was much more important in traditional than in modern analyses. In our study, the modern approach provided different results according to the type of data: satisfactory when using pseudolandmarks (outlines), less satisfactory when using landmarks. The different reclassification methods provided approximately similar scores, with a special mention to the neural network, which allowed improvements in accuracy by combining data from both morphometric approaches.

Conclusion

We conclude that morphometrics, whether traditional or modern, represent a valuable tool to assist in Fasciola species recognition. The general level of accuracy is comparable among the various methods, but their demands on skills and time differ. Based on the outline method, our study could provide the first description of the shape differences between species, highlighting the more globular contours of the intermediate forms.

Introduction

Fasciola hepatica Linnaeus, 1758 and F. gigantica Cobbold, 1855 are well-known as parasites that cause fascioliasis (Mas-Coma, Bargues & Valero, 2005). The parasites belong to the family Fasciolidae Railliet, 1895, subfamily Fasciolinae Stiles & Hassall, 1898. The adult worm infects the liver of various species of mammals, particularly livestock that consume fresh plant material contaminated with infective-stage metacercaria (Mas-Coma, Bargues & Valero, 2018). Climatic changes seem to have exacerbated the problem by expanding the geographic distribution of such parasites (Mas-Coma, Valero & Bargues, 2008; Mas-Coma, Valero & Bargues, 2009a). Typically, F. hepatica is reported worldwide, while F. gigantica occurs in a narrower range across Asia and Africa. The geographic distribution of these two parasites can overlap, particularly in the equatorial zone. The hybrid forms that can result between F. hepatica and F. gigantica are often reported as Fasciola “intermediate forms” (Periago et al., 2008; Mas-Coma, Valero & Bargues, 2009a; Mas-Coma, Valero & Bargues, 2009b). Like other countries in the tropical and subtropical areas, Thailand has reported Fasciola infections in both humans and livestock (Mas-Coma, Valero & Bargues, 2009a; Buranasin & Harinasuta, 1970; Srihakim & Pholpark, 1991; Aroonroch et al., 2006; Kanoksil et al., 2006; Tippawangkosol et al., 2017; Wannasan et al., 2014).

Up until now, traditional morphometrics, which is based on linear measurements between anatomical landmarks, has been used to help in the identification of Fasciola species (Valero, Panova & Mas-Coma, 2005). Many factors have been noted which can modify the accuracy of phenotypical classification done in this way, including: (1) lymnaeid-host specificity, (2) bias arising from different types of definitive hosts, the technical bias for worm fixing, (3) the computerized image analysis system used, (4) the standardized methodology used for measurements, and (5) the effect of allometric growth (Valero, Panova & Mas-Coma, 2005; Periago et al., 2006). Size and some other variables or ratios that have been erroneously considered as a proxy for “shape” (body roundness (BR = BP2/4 πBA; BP = body perimeter; and BA = body area) could discriminate between adult F. hepatica, F. gigantica, and Fasciola intermediate forms collected from different countries in Asia (Periago et al., 2006), as well as being used to examine intraspecific diversity (Periago et al., 2008). A molecular identification technique was developed to discriminate between fasciolid adult worms based on ribosomal internal transcribed spacers (ITS) 1 and 2 (Lin et al., 2007; Lotfy et al., 2008). According to Amer et al. (2011), the identification of fasciolids requires at least ITS2 and mitochondrial cox1 or nad1 sequences. Itagaki et al. (2011) suggested that ITS1 sequences, which do not show variation in Europe or Oceania, are nonetheless useful molecular markers for discriminating F. hepatica, F. gigantica, and their hybrids occurring in Asia (Amer et al., 2011).

In Thailand, Fasciola intermediate forms were found in the livers of cattle in slaughterhouses and identified as hybrids because of the heterosequences of nuclear ribosomal ITS1 and ITS2 (Chaichanasak et al., 2012; Siribat et al., 2018). For molecular discrimination between F. gigantica, F. hepatica, and Fasciola intermediate forms, ITS1 has been noted as the most effective DNA marker (Chaichanasak et al., 2012). Fasciola intermediate forms present no qualitative morphological attributes that allow clear identification. Alongside molecular discrimination techniques, traditional morphometrics has been suggested as a relatively simple and effective approach for species identification (Periago et al., 2006; Valero, Marcos & Mas-Coma, 1996; Valero, Panova & Mas-Coma, 2005; Ashrafi et al., 2006; Afshan et al., 2014). However, the dimensions of Fasciola intermediate forms may overlap with those of both F. gigantica and F. hepatica (Periago et al., 2006; Srimuzipo et al., 2000).

Even when using many characters and multivariate statistical tools, the risk of using a morphometrics method is that linear distance measurements mainly provide information relating to size and provide limited information about shape. Previous studies inaccurately labeled as “shape” some variables which are actually another estimation of size, such as body roundness (BR = BP2/4πBA), body perimeter (BP), and body area (BA) (Periago et al., 2006; Mas-Coma, Valero & Bargues, 2009b; Valero et al., 2018). These characters are secondary constructions from linear measurements and produce variables redundant with size (Burnaby, 1966).

A more convenient approach has recently been applied which enables variations in shape to be estimated from linear measurements (Valero et al., 2018). However, there are various techniques available that can be used for this. A comprehensive review was recently published distinguishing two “schools” via which it is possible to approach the allometric concept: the Huxley–Jolicoeur and the Gould–Mosimann schools (Klingenberg, 2016). The former, initially developed for bivariate analyses, considers morphological form as a single unified feature (form as both size and shape), emphasizing the covariation among traits as a consequence of variation in size. The latter, initially developed for multivariate morphometric measurements, separates two estimates: an estimate of global size and a multivariate estimate of shape (Mosimann, 1970; Darroch & Mosimann, 1985). According to Klingenberg (2016), landmark-based geometric morphometrics represents a straightforward extension to this second method. Its concept may be naturally extended to the outline-based approach, which also separates a global estimator of size and a multivariate estimate of shape (Rohlf & Archie, 1984; Lestrel, 1989). Instead of using distances between anatomical landmarks, the modern methods use their coordinates in a common system of axes. One of the clear improvements introduced by the modern approach is that it enables a visual representation of shape (Rohlf & Marcus, 1993). Visualization of changes in shape are frequently presented using D’Arcy Thompson-like plots, or deformation grids (Bookstein, 1991). Deformation grids help to visualize the shifts in aligned coordinates from one shape to another. The grid is a visual aid and can even be amplified to improve the visibility of subtle changes. It can be omitted when these changes are clearly visible, as is frequently the case between species, without the need for any amplification artefacts.

The use of coordinates usually applies to animal species presenting a solid structure on their body. However, this method has also been applied to soft structures. For example, when combined with photogrammetric techniques, it can reveal variations in the shape of rodent sperm-cell heads and the diversity of these shape variations among species (Varea Sánchez, Bastir & Roldan, 2013).

In our study, we used recommended traditional morphometric techniques and, to preserve the parallelism of shape extraction from the data, we applied the Darroch & Mosimann technique (1985) to the linear measurements, which produces “log-shape ratios” as estimates of shape.

We compared the results obtained by traditional techniques with those obtained from two methods based on the coordinates of anatomical points: the landmark-based (Rohlf, 1990) and the outline-based (Kuhl & Giardina, 1982; Lestrel, 1997) approaches. These two approaches are conceptually different. The landmark-based approach uses anatomical points called “landmarks”. These points are small biological structures, as small as a point at the requisite scale. They are considered to be homologous in the sense that they are relocatable points from one individual to another. The outline-based approach also uses an ordered set of discrete point coordinates; these are superficially similar to landmark data, but they are conceptually quite different. They are of a different nature to true landmarks because comparability between them is not expected separately, but from the structure they describe. The outline-based method is generally restricted to closed contours (outlines) where landmarks are lacking. In this approach, the contour is described by points that we call “pseudolandmarks” (Dujardin, 2017). There is no need for these pseudolandmarks to be equally spaced, and there is no need to have the same number of them from one individual to another (Rohlf, 1990; Dujardin et al., 2014).

Pseudolandmarks should not be confused with “semilandmarks”. Semilandmarks are landmarks that lack one degree of freedom because they depend on other landmarks. They are generally used to compare open curves between other landmarks, and, most importantly, they are processed as true landmarks by Procrustes superposition, combined with a sliding procedure to minimize total bending energy or to minimize the Procrustes distances to the consensus (Dujardin, 2019).

Outline data are processed in a very different way, either by elliptic Fourier analysis (Kuhl & Giardina, 1982) or, less frequently, by eigenshape analysis (Lohmann, 1983). Briefly, the observed contour is decomposed in terms of sine and cosine curves of successive frequencies called harmonics, with each harmonic being described by four coefficients. These coefficients are normalized by the semi-grand axis of the first harmonic ellipse. The latter may be used as an estimate of global size, as well as the root-squared area of this same starting ellipse, the root-squared area of the digitized contour, or its perimeter. We explored the capacity of these different approaches to discriminate between F. hepatica, F. gigantica, and Fasciola intermediate forms.

To ensure correct evaluation of the methods, the samples collected from Thailand were also identified using molecular analysis. In addition to the Thailand samples, nine F. hepatica strains from the UK were provided and used to give reference data: they were subjected to morphometric analyses only.

As far as we know, morphometrics applied to the Fasciola identification always relied on traditional morphometric techniques. In this study, we tried to answer the following question: could modern morphometric techniques bring any improvement over traditional ones in terms of accuracy and/or simplicity?

Materials & methods

Sample collection

Ninety adult worms of Fasciola spp. were collected from the livers of cattle slaughtered in local slaughterhouses in Lamphun, Mae Hong Son, Tak, and Pratum Thani Provinces, Thailand. The cattle in the Pratum Thani slaughterhouse were from Kanchanaburi Province, Thailand. All cattle came from local farms in Thailand. Infected livers were stored on ice and transferred to the laboratory at the Department of Helminthology, Faculty of Tropical Medicine, Mahidol University, within 12 h and within 1 h for the samples from Pratum Thani province. Adult Fasciola worms were gently separated from the liver tissue. Each individual worm was fixed between two transparent glass slides containing two pieces of paper (0.4 mm) and the two slides were gently held in place using a rubber band. The specimens were placed in normal saline for 1 h. The lateral part of the body of each adult worm was then cut away and stored individually in absolute alcohol at −20 °C until being used for the molecular study. The remaining section of each worm’s body was washed in distilled water for 5 min to relax the structure before being placed individually between glass slides. The fixed adult worms were tentatively identified under a stereomicroscope using the taxonomic criteria of primary morphological characters, including cephalic cone, body length, and body width (Periago et al., 2008; Valero, Marcos & Mas-Coma, 1996). The specimens were fixed in 70% ethanol and stored at 4 °C for staining.

In addition to our field specimens, nine F. hepatica specimens were obtained from the Liverpool School of Tropical Medicine (UK). These specimens were used for comparison and subjected to both traditional and geometric morphometric examination. All specimens of F. hepatica, which had sheep as their definitive host, were received as permanently mounted slides and deposited in our laboratory in 1993. They were stained with Semichon’s acid-carmine and mounted with Canada balsam.

To make sure that our relatively small samples did not generate outlier values, we compared graphically the observed means obtained using our material to values already published for the same hosts in different countries (Valero, Panova & Mas-Coma, 2005; Halakou et al., 2017; Shafiei et al., 2014; Iyiola et al., 2018).

Molecular identification by PCR-RFLP

A small piece of adult Fasciola worm was cut from the lateral part of the body. Genomic DNA was extracted from each individual using a Tissue Genomic DNA Mini Kit (Geneaid, Taipei, Taiwan), according to the manufacturer’s protocol. The PCR amplicons of the ribosomal ITS1 and ITS2 regions were amplified using the primers ITS1-F and ITS1-R for ITS1 and 3S and BD2 for ITS2 (Siribat et al., 2018; Ichikawa & Itagaki, 2010; Huang et al., 2004). We conducted PCR amplification of the ITS1 and ITS2 regions three times for each sample. Each PCR reaction was performed in 50 mL final volume composed of 1 × TopTaq Master Mix Kit (Qiagen, Hilden, Germany) (1U TopTaq polymerase, 1.5 mM MgCl2, and TopTaq polymerase buffer), 20 pmol of each primer, and 10 ng/ µL genomic DNA template. The PCR cycles consisted of initial denaturation at 95 °C for 3 min followed by 29 cycles of denaturation at 95 °C for 45 s, annealing at 55 °C for 30 s, and extension at 72 °C for 60 s. The final extension of each PCR reaction was conducted at 72 °C for 8 min.

After purifying the PCR products using the ethanol precipitation method, 500 ng of purified PCR products from each sample was digested with restriction enzymes Rsa I and Nla III for ITS1 and ITS2, respectively (New England BioLab Inc., Massachusetts, USA) at 37 °C for 150 min. The enzymatic digestion was inactivated by heating at 80 °C for 20 min. The final digested PCR amplicons were run on a 2% agarose gel at 50 V for 180 min. The band patterns were visualized using a UV transilluminator. The PCR-RFLP figure was photographed with Gel Documentation (G-Box (HR), Syngene, UK).

Parasite staining

The unfolded worms were stained overnight with Semichon’s acid-carmine, then de-stained with 1% hydrochloric acid for 5 min. The stained specimens were dehydrated in an ethanol dilution series of 70%, 80%, 90% (twice), 95% (twice), and finally in absolute ethanol (twice). Each dehydration step took 30 min. The stained specimens were performed using 1:1 xylene and absolute ethanol for 30 min before being transferred to pure xylene for 30 min; by doing it twice. Finally, the specimens were mounted with Canada balsam (Waikagul & Thaenkham, 2014).

After staining, all specimens were photographed using a Nikon DS-Ri1 SIGHT digital camera connected to a Nikon AZ 100 M stereomicroscope (Nikon Corp., Tokyo, Japan), and the scale was indicated on each photograph. An initial classification was performed based on a visual examination of the specimen’s external morphology. The results of these identifications were compared with the outcomes of the PCR-RFLP molecular identification using ITS1 and ITS2 as the molecular markers (Table 1). After that, ImageJ software (freely available at https://imagej.nih.gov/ij/) was used to measure the 20 morphological characters used in traditional morphometric analyses (Table 2).

Table 1 Visual identification versus molecular typing of field specimen.

The visual examination of external morphology could not recognize any of the 16 intermediate forms, most of them (14) wrongly identified as F. hepatica. A minor part of F. gigantica (13) was misidentified also with F. hepatica.

Species	External morphology	PCR-RFLP	
F. gigantica	63	74	
Fasciola intermediate forms	0	16	
F. hepatica	27	0	

Table 2 Traditional morphometric analyses of Fasciola species.

Character	Fgn = 17	Fifn = 10	Fhn = 9	P	
	M	Min-Max	SD	M	Min-Max	SD	M	Min-Max	SD	Fg-Fif	Fg-Fh	Fif -Fh	
1. BL	29.9	20.8–42.1	6.7	16.0	10.9–23.2	3.6	28.2	23.8–33.3	3.1	0.000	0.787	0.000	
2. BW	7.4	4.8–10.9	1.9	8.6	5.9–12.8	2.2	10.2	8.2–12.9	1.5	0.141	0.001	0.107	
3. BWOv	4.4	3.3–5.9	0.7	5.1	3.3–7.7	1.3	5.1	4.2–5.6	0.5	0.113	0.063	0.746	
4. BP	65.4	46.3–88.2	13.6	37.5	29.2–57.5	9.0	63.3	54.6–74.8	6.6	0.000	0.900	0.000	
5. CL	2.9	2.4–3.7	0.4	1.6	1.2–2.1	0.3	3.4	2.6–3.9	0.4	0.000	0.241	0.000	
6. CW	3.3	2.4–4.1	0.5	2.4	1.80–3.35	0.5	4.1	3.4–4.6	0.4	0.003	0.021	0.000	
7. OSmax	0.8	0.5–1.0	0.1	0.6	0.5–0.7	0.1	0.8	0.5–1.1	0.2	0.001	0.861	0.001	
8. OSmin	0.5	0.3–0.7	0.1	0.4	0.3–0.6	0.1	0.6	0.3–0.8	0.2	0.399	0.089	0.023	
9. VSmax	1.3	0.9–1.6	0.2	0.9	0.7–1.1	0.1	1.5	0.9–1.7	0.3	0.000	0.106	0.000	
10. VSmin	1.3	1.0–1.6	0.2	0.8	0.5–0.9	0.2	1.5	1.1–1.7	0.2	0.000	0.239	0.000	
11. A-VS	2.2	1.5–2.9	0.1	1.5	1.1–2.3	0.3	2.9	2.6–3.5	0.3	0.000	0.032	0.000	
12. OS-VS	1.6	0.9–2.5	0.4	1.0	0.6–1.6	0.3	2.3	1.9–3.1	0.4	0.003	0.031	0.000	
13. VS-Vit	14.4	9.8–21.8	4.2	8.1	6.0–12.1	2.0	17.0	12.8–25.9	4.4	0.000	0.264	0.000	
14. Vit-P	13.5	5.8–33.4	7.1	4.1	1.3–8.1	1.7	7.7	5.8–10.8	1.4	0.000	0.059	0.014	
15. VS-P	24.3	14.1–37.8	7.3	12.3	8.8–20.0	3.3	23.7	19.3–29.0	4.3	0.000	0.946	0.000	
16. PhL	0.6	0.4–0.8	0.1	0.4	0.2–0.5	0.1	0.7	0.6–0.9	0.1	0.000	0.265	0.000	
17. PhW	0.5	0.3–0.8	0.1	0.3	0.2–0.5	0.8	0.5	0.3– 0.6	0.1	0.000	0.794	0.000	
18. TL	12.6	8.3–19.1	3.7	7.4	5.7–11.0	1.7	13.7	10.7–19.7	2.8	0.000	0.499	0.000	
19. TW	4.2	2.5–7.5	4.2	5.3	3.7–8.1	1.5	6.2	4.7–7.6	0.9	0.049	0.000	0.239	
20. TP	31.6	22.2–43.1	7.6	25.0	17.8–33.2	6.1	35.9	30.6–46.1	4.7	0.019	0.163	0.000	
Notes.

Fg Fasciola gigantica

Fh Fasciola hepatica

Fif Fasciola intermediate forms

M mean

n sample size

Min minimum

Max maximum;

SD standard deviation

BL Body length

BW Body width

BWOv BW at ovary level

BP Body perimeter

CL Cone length

CW Cone width

Osmax Oral sucker maximum diameter

Osmin Oral sucker minimum diameter

Vsmax Ventral sucker maximum diameter

Vsmin Ventral sucker minimum diameter

A-VS Distance between anterior end of body and VS

OS-VS Distance between suckers

VS-Vit Distance between VS and union of vitelline glands

Vit-P Distance between Vit and posterior end of body

VS-P Distance between VS and posterior end of body

PhL Pharynx length

PhW Pharynx width

TL Testicular space length

TW Testicular space width

TP Testicular space perimeter

Data processing

Among the 74 and 16 genetically identified F. gigantica and Fasciola intermediates, respectively, we selected the less damaged specimens that constituted the best samples for the morphometric studies: 17 F. gigantica and 10 Fasciola intermediate form specimens (from this study). In addition, nine F. hepatica (from Liverpool School of Tropical Medicine, UK) were included in the material.

Four types of morphometric data were separately analyzed: (i) log-transformed linear measurements (LN), (ii) log-shape ratios (LSR), (iii) outlines data (OD), and (iv) landmarks data (LD). For the artificial neural network-based classification (ANN, see below), we also tested the combination of LN with OD.

Software

ImageJ image analysis software was used to measure linear dimensions. Landmarks and pseudolandmarks (outlines) were digitized using the CLIC package (https://www.xyom-clic.eu). Subsequent analyses and graphical outputs were performed using both the CLIC package and the recently available online XYOM software (https://xyom.io/) (Dujardin & Dujardin, 2019). The latter allowed some specific analyses to be performed, including the maximum likelihood (MxL) and the ANN methods of classification (see below).

Traditional morphometrics analyses

Twenty characters were measured using a computerized image analysis system, as described by Valero and colleagues (Valero, Marcos & Mas-Coma, 1996; Valero, Panova & Mas-Coma, 2005). We compared graphically our data with previously published ones, showing the departures of observed means (between the published means and ours) from the total means (Fig. S1).

Univariate morphometric comparisons used non-parametric techniques based on 1,000 random permutations of the data between groups. The linear measurements (LN) were log-transformed for the multivariate analyses and validated classification. In addition, the eigenvectors of PC1 and PC2 were used to extract the variables that made the largest contribution (>25%), resulting in a total of 12 variables instead of 20. These were BL, BW, BP, CL, Vsmin, A-VS, OS-VS, VS-Vit, Vit-P, VS-P, TW, and TP (see Table 2). Log-linear measurements were then subjected to the Darroch & Mosimann (1985) procedure to extract an estimate of global size (log-size) and an estimate of shape variables (log-shape ratios, LSR). The influence of size on interspecific discrimination by LSR was estimated by regressing on log-size the discriminant factors.

Outline data (OD)

The contour of each specimen was manually digitized (see Fig. 1A), and the resulting coordinates were subjected to an elliptic Fourier analysis. The “outline data” (OD) used as the input for validated classification were the normalized elliptic Fourier coefficients (NEF). Their numbers were restricted to an average set giving a power of at least 99% (n = 30). Their first eight principal components were used directly as input for the validated classification tests. An estimate of the allometric residue contained in shape variables was performed by regressing the principal components of the NEF on the square root of the area delimited by the contours. The non-parametric estimation of the statistical significance of this multivariate regression followed the procedure of Good (2000). On the other hand, the influence of the global size on interspecific discrimination itself was estimated after regressing the first discriminant factor on size. For the artificial neural network technique (ANN, see below), the OD were also combined with different numbers (4, 6, 8, 10, and 12) of LN.

Figure 1 Geometric morphometric analysis of Fasciola adult worms.

(A) Contour digitized on Fasciola spp. body for outline-based geometric morphometrics analysis. (B) Position of five landmarks digitized on Fasciola spp. body for landmark-based geometric morphometric analysis.

Landmarks data (LD)

The following anatomical landmarks (LM) were digitized: LM1 = left-side of the cephalic cone, LM2 = left-side of the oral sucker, LM3 = right-side of the oral sucker, LM4 = right-side of the cephalic cone, and LM5 = end of the testis (see Fig. 1B). The five landmarks were subjected to a generalized Procrustes analysis, and the resulting aligned individuals were orthogonally projected onto the Euclidean space tangent to the consensus of forms (Rohlf & Slice, 1990). The principal components (PC) of these data (called “Procrustes residuals”, “tangent space variables”, or “orthogonal projections”) were referred to here as “landmarks data” (LD) and used as the input for validated classification. The influence of size on interspecific discrimination by LD was estimated by regressing the discriminant factors on the centroid size.

Measurement error due to imaging and digitizing

The precision of our digitization was evaluated for both landmarks and outlines. For size we used the method recommended by Arnqvist & Mårtensson (1998), and for shape we used the Procrustes analysis of variance (Procrustes ANOVA) (Klingenberg & McIntyre, 1998).

Validated classification

To assess how our results could be generalized to an independent data set, each individual was iteratively removed and its assignation computed following the analysis of the remaining data (the “leave-one-out” method, also known as “validated reclassification”) (see Data S1). The analyses of the remaining data used three different statistical tests: a distance-based (Dist) method, a maximum likelihood (MxL) method, and an ANN method.

In the first method (Dist), the Mahalanobis distance was computed between the testing set (one individual) and the average of each training group (the remaining data), and the tested individual was assigned to the closest group. In the second method (MxL), the testing set (one individual) was assigned to the group it was thought most likely to belong to assuming a normal distribution and independence between characters (Polly & Head, 2004; Dujardin, Dujardin & Kaba, 2017). For the Dist and MxL methods we tried to avoid using a number of input variables that would exceed the number of individuals in the smallest sample. There are various methods to deal with multidimensionality and we selected the simplest one: the set of first PCs lower than the number of individuals in the smallest sample. Note that for the analyses related to the use of five anatomical landmarks, the total number of final shape variables was actually six (two-times the number of LM minus 4 in the case of two-dimensional analyses).

The ANN method made use of a simple multilayer perceptron with a back propagation algorithm (Rumelhart, Hinton & Williams, 1988), a variant of the original perceptron model (Rosenblatt, 1958) that is not, however, restricted to linear models. The method has been recently applied to morphometric data, including both OD (Mancuso, 1999) and LD (Dobigny, Baylac & Denys, 2002; Baylac, Villemant & Simbolotti, 2003; Van Bocxlaer & Schultheiß, 2010; Lorenz, Ferraudo & Suesdek, 2015; Soda, Slice & Naylor, 2017). We used the multilayer perceptron program written in JavaScript, which is available at https://www.npmjs.com/package/mlp.

Following a process of trial-and-error, a single (hidden) layer composed of three neurons provided the best results, irrespective of the type of input data. We used as the input the total number of variables instead of a subset of their PCs. Inputs comprised the morphometric variables (LN, LSR, OD, or LD, see above), while the final outputs were the probabilities of inclusion in each group (in a Bayesian sense). Forms were then classified to the species for which their probability of inclusion was the greatest (Soda, Slice & Naylor, 2017). The “weights”, i.e., the coefficients allowing the outputs to be computed, were applied to the individuals being tested to compute categories and suggest a classification. Thus, each individual was tentatively assigned to its putative species after computing the weights from the remaining data. For a single individual to identify, the remaining data were subjected to a large number of iterations to reduce the classification error to a minimum (less than 25%). All specimens were separately classified 30 times, then an average classification and its standard error were computed (see Data S1).

To make the most of the less stringent statistical assumptions of the ANN method with regard to the statistical distribution of the input data (Ripley, 1996), we explored its capacity to deal with different metrics combined into a single matrix. Thus, we built a matrix with four LN and made another (normalized) matrix combining them with the OD. We performed a validated reclassification on each matrix and then compared the scores with the correct identification. We repeated this comparison successively using 6, 8, 10, and finally 12 LN, combined with the same set of OD.

Results

External morphological classification

The rough identification of our 90 field specimens did not detect the presence of Fasciola intermediate forms despite them being present, as determined by the molecular analyses (see next paragraph). Moreover, the rough identification wrongly assigned 30% of the specimens as F. hepatica species, despite the complete absence of this species, as revealed by the molecular data (Table 1).

Molecular identification

The PCR amplicons of the ITS1 and ITS2 regions (680 and 550 bp) from 90 adult worms of Fasciola spp. were digested by Rsa I and Nla III, respectively, and provided species-specific band patterns that enable the three Fasciola species to be distinguished (Figs. S2, S3). The results showed that 74 worms were classified as F. gigantica and 16 worms were Fasciola intermediate forms. F. hepatica was not present in the samples collected in the field in Thailand (Table 1).

Traditional morphometrics (univariate analyses)

Six out of 20 characters showed significant differences between the three species: CL, CW, VSmax, VSmin, A-VS, and OS-VS (Table 2). Between Fasciola intermediate forms and the other two species, BL, BP, OSmax, VS-Vit, VS-P, PhL, PhW, and TL were statistically significantly different. The remaining variables appeared to be similar among the three species (Table 2).

Traditional morphometrics (multivariate analyses)

Based on the complete set of 20 linear measurements, the validated classification produced excellent scores of correct species attribution, ranging from 89% to 92% according to the reclassification method (Table 3, column “20LN”). In order to tentatively reduce the number of characters, we selected a subset of contributing variables. We did not select them according to univariate statistical differences between groups but according to their contribution to the first and second principal components. This selection allowed us to restrict the set of LN from 20 to 12 variables (BL, BW, BP, CL, Vsmin, A-VS, OS-VS, VS-Vit, Vit-P, VS-P, TL, and TW). The exclusion of poorly contributing variables was beneficial and substantially improved the validated reclassification score from 88% to 97% (Table 3, column “12LN”). The shape variables derived from LN (either LSR from 20LN or LSR from 12LN) produced very similar scores as log-transformed raw variables (Table 3).

Table 3 Validated classifications according to three methods and three morphometric approaches (Traditional, Outlines and Landmarks).

Percentages of correct classification according to data sets and classification methods.

Methods	Samples/ Variables	Traditional	Outlines	Landmarks	
		20 LN	LSR (20 LN)	12 LN	LSR (12 LN)	30 OD	6 LD	
ANN	17 Fg	13.63 [0.85]	13.5 [1.01]	13.47 [1.04]	14.17 [1.05]	14.96 [0.69]	10.81[2]	
	9 Fh	8.4 [0.67]	7.6 [0.72]	8.43 [0.68]	7.73 [0.69]	7.13 [0.85]	5.65 [1.21]	
	10 Fif	9.97 [0.18]	9.93 [0.25]	9.83 [0.46]	8.43 [0.73]	9.96 [0.20]	8.33 [0.95]	
	36	89% [2.8]	86% [3.5]	88% [3.5]	84% [2.8]	89% [3.5]	68% [5.6]	
MxL (PC)	17 Fg	16	15	17	15	14	9	
	9 Fh	7	7	7	7	8	6	
	10 Fif	9	10	8	9	10	6	
	36	89% (3)	89% (4)	89% (4)	86% (3)	89% (4)	58% (3)	
Dist (PC)	17 Fg	15	15	17	17	16	13	
	9 Fh	8	8	8	9	8	5	
	10 Fif	10	10	10	9	10	7	
	36	92%	92%	97%	97%	94%	69%	
Notes.

Fg Fasciola gigantica

Fh Fasciola hepatica

Fif Fasciola intermediate forms

LN natural logarithms

LSR Log-Shape-Ratios

OD outline data, here Normalized Elliptic Fourier coefficients

LD Procrustes residuals (orthogonal projections)

ANN artificial neural network. For ANN, the results are presented as averages over 30 sessions and, between square brackets, the standard deviation

MxL maximum likelihood validated reclassification: between brackets, the number of PC giving the best reclassification score; the Dist (PC) the Mahalanobis distance-based validated method used the 8 first principal components of the variables LN and OD, and the 6 PC of ORP

Allometry

The influence of size on LSR-based discrimination was high: 61% on the first discriminant factor (DF1) and 63% on the second (DF2).

Repeatability

The precision of the global size estimation was almost perfect (99%) when using both the landmark and outline techniques. For shape, the precision was slightly lower when using the landmarks technique (93%) compared with the outlines technique (98%).

Global size

The global size of the parasites, either the body perimeter as computed from OD, the square-root area within the contour (Fig. 2A; Table 4A) or the centroid size as computed from LD (Fig. 2B, Table 4B) differed significantly between the intermediate forms and the other two species. These other two species, F. hepatica and F. gigantica, did not show statistically significant differences in global sizes (Table 5). Fasciola intermediate forms had the smallest global size, both in terms of centroid size and the perimeter or square-root area of the contour.

Figure 2 Global size of Fasciola. adult worms.

(A) Quantile boxes showing body perimeter variation obtained after outline-based geometric morphometric analysis. Each box shows the group median that separates the 25th and 75th quartiles. Dots on the left represent individuals. (B) Variation of body centroid size of Fasciola spp. from landmark-based GM analysis. Each box shows the group median that separates the 25th and 75th quartiles. Dots on the left represent individuals.

Table 4 Global sizes of Fasciola spp.

Species	n	Mean	Min	Max	SD	
4a. Body perimeter and Square-root area of Fasciola spp.	
	Body Perimeter	
Fg	17	65.44	45.22	88.47	13.96	
Fh	9	62.25	55.55	75.40	6.88	
Fif	10	38.67	29.05	56.83	9.14	
	Square-root area	
Fg	17	13.5	9.04	17.89	2.54	
Fh	9	14.3	10.64	14.68	1.76	
Fif	10	9.8	5.20	10.40	2.33	
4b. Centroid size of Fasciola spp.	
Fg	17	17.44	11.83	28.33	4.98	
Fh	9	17.16	14.48	22.00	2.52	
Fif	10	9.45	6.93	13.36	2.31	
Notes.

Fg Fasciola gigantica

Fif Fasciola intermediate forms

Fh Fasciola hepatica

n number of specimen

Min minimum

Max maximum

SD Standard deviation

Perimeter and square-root area as computed by the outline-based technique, and centroid size by the Generalized Procrustes Analysis (landmark-based technique).

Table 5 Non-parametric comparisons of global size estimations, 1,000 permutations (P-values).

Species	Centroid size	BP / Square-root area	
F. gigantica / F. hepatica	P=0.909	P = 0.650∕P = 0.555	
F. gigantica / F. intermediate forms	P < 0.001	P < 0.001/P = 0.003	
F. intermediate forms / F. hepatica	P < 0.001	P < 0.004/P < 0.001	
Notes.

P percentage of values equal or higher than the observed differences of mean values between species after 1,000 random permutations. It is the empirical risk of error saying that the differences are significant

BP body perimeter

Outline-based analyses

Shape variation among Fasciola spp. according to the outline-based analysis

Direct visualization, without amplification, of contour differences suggested Fasciola intermediate forms possess a more globular contour (Fig. 3A). The factor maps of the discriminant analyses (Fig. 3B) showed non-overlapping distributions. Accordingly, satisfactory validated classification scores were obtained, ranging from 89% to 94% (Table 3, column “OD”).

Figure 3 Shape variation of Fasciola. spp. based on the outline-based analysis.

(A) Mean shape of whole body contour of F. gigantica, F. hepatica, and Fasciola intermediate forms. (B) Outline-based discriminant analysis. Factor map of the two discriminant factor (DFs) derived from shape variables for Fasciola species. Each point represents an individual. The horizontal axis is the first DF; the vertical axis is the second DF; their cumulated contributions reach 100% of the total variation.

Allometry

The contribution of global size variation to the first shape-derived discriminant factor derived from OD was 23%, while it was 6% for the second one. According to the non-parametric method used to evaluate the multivariate regression significance between size and shape, the allometric residue contained in OD shape variables was significant (p < 0.001).

Landmark-based analyses

The mean specimens aligned according to the five anatomical landmarks used here showed some separation between the three forms, with a seemingly more critical difference for Fasciola intermediate forms (Fig. 4A), as for the OD (Fig. 3). The factor map of the discriminant analysis showed that Fasciola intermediate forms could diverge from F. gigantica and F. hepatica (Fig. 4B). However, some overlapping was observed, and the validated classification scores based on landmark data were not satisfactory, ranging from 58% to a maximum of 69% (Table 3, column “LD”).

Figure 4 Shape variation of Fasciola. spp. based on landmark-based analyses.

(A) Average positions of anatomical landmarks (dots) after Procrustes superposition, representing the mean shapes of F. gigantica, F. hepatica, and of Fasciola intermediate form. (B) Landmark-based discriminant analysis. Factor map of the two discriminant factors (DFs) derived from final shape variables of F. gigantica, F. hepatica, and Fasciola intermediate forms. Each point represents an individual. The horizontal axis is the first DF; the vertical axis is the second DF, their cumulated contributions reach 100% of the total variation.

Statistical significance of shape variation

The pairwise Mahalanobis distances based on shape, for both landmark- and outline-based shapes, were highly significant (p < 0.001) between Fasciola intermediate forms and F. gigantica or F. hepatica. Between F. gigantica and F. hepatica, the distance was smaller, albeit almost significant (p ∼ 0.06) (Table 6).

Allometry

The first and second discriminant factors derived from the Procrustes residuals were still under the influence of size (42% and 4%, respectively) after regression on centroid size, which was more than for the outline-based analysis (23% and 6%, respectively) but much less than for the traditional method (61% and 63%, respectively).

Combination of log-transformed linear measurements and OD

The ANN performance was significantly improved when traditional linear measurements (LN), even a small number of them, were combined with outline shape descriptors (OD). This performance could be appreciated by comparing the ANN accuracy when applied to these data either separately or combined (Table 7). For instance, with ten traditional measurements as the input, the ANN performance achieved 88% of correct identification, with OD only it reached 89% (Table 3), whereas by combining both types of data the total score jumped to 94% (Table 7).

Table 6 Mahalanobis distance and P- value for outline-based and landmark-based GM analyses.

Species	Outline-based GM	Landmark-based GM	
	Fg	Fh	Fif	Fg	Fh	Fif	
Fg	0.00			0.00			
Fh	2.28 (p = 0.108)	0.00		1.68 (p = 0.063)	0.00		
Fif	4.42 (p < 0.001)	4.23 (p = 0.001)	0.00	2.85 (p < 0.001)	2.86 (p < 0.001)	0.00	
Notes.

Fg Fasciola gigantica

Fg Fasciola hepatica

Fif Fasciola intermediate forms

Table 7 Scores of validated classification using as input for the multilayer perceptron either log-transformed linear measurements (LN) alone, or a combination of LN with outline data.

Percentages are the average of 30 ANN classification sessions (see Data S1), with the standard deviation between square brackets.

	LN	LN + OD	
	average	stdev	average	stdev	
4 LN	73%	[4.8]	88%	[4.0]	
BL, VS-Vit, Vit-P, VS-P	
6 LN	78%	[4.2]	88%	[4.0]	
4LN above + OS-VS, A-VS	
8 LN	81%	[4.5]	88%	[5.3]	
6LN above + OS-VS, CL, TL, BP	
10 LN	88%	[3.5]	94%	[3.7]	
8 LN above + BW, TW	
12 LN	88%	[3.5]	96%	[3.0]	
10 LN above + VSmin, A-VS	
Notes.

LN log-transformed linear measurements

OD outline data, i.e., normalized elliptic Fourier coefficients

The first row may be read as follows: with only four log-transformed linear measurements (4LN, first column), 78% of correct species assignment was obtained after validated classification (second column), 88% when these data were combined with outline data (third column).

Discussion

The specimens in this study were processed using the same treatment method. If some deformation occurred as a result of this treatment, it is reasonable to expect that this would be the same for each species. The average values of linear measurements of 12 selected morphological characters of each Fasciola sp. were graphically compared with previously published values from the same parasites and the same hosts (see Fig. S1). This comparison showed that our samples were in the range of average values and suggested that the small number of specimens did not result in any sample bias.

To reduce the effects of growth on the phenotype, only fully mature stages were selected from each molecularly defined group. However, in addition to possible growth heterogeneity, other divergent influences could have affected phenotypic variations in our total sample, such as different geographic origins, microhabitats, host, and seasonality. We considered that, within each species or group, these environmental effects would probably have had much more of an impact on size than on shape. We used the geometric morphometrics approach to tentatively set apart these size effects from the morphometric data, extracting the remaining shape variables and using them exclusively to describe the groups distinguished by molecular data. The allometric residue that possibly remained in the shape variables was not removed because our comparisons concerned species rather than intraspecific groups. Species may differ by their allometric growth, which becomes part of their evolutionary divergence (Klingenberg, 2016).

Our tentative removal of the size effect from the linear measurements (either 20 LN or 12 LN) did not change the global classification scores as obtained by raw measurements, probably because it could not perform a consistent removal of size, as indicated by our allometric analysis. Size remained dominant in the LSR (∼61%) relative to the PR (∼41%) and OD (∼24%). Another inconvenience of LSR is that they do not allow a visual representation of shape changes between taxa (Rohlf & Marcus, 1993). Methods based on coordinates data allow such visual representation (see Fig. 2D).

Our samples represented limited geographic isolates and, because of the low sample sizes, could suffer from low representativity. Moreover, the number of variables could be greater than the number of individuals in our smallest sample (nine F. hepatica). We followed the “rule” for a multivariate analysis, especially discriminant analysis, which is to use a number of variables that does not exceed the number of individuals in the smallest sample. There are various methods available to deal with multidimensionality. This specific number of PC scores was selected so as to use a number of variables that does not exceed the number of individuals in the smallest sample. This procedure allowed multidimensionality to be reduced where necessary; this did not make our samples more representative, but at least the information they provided was not over-interpreted.

Since linear measurements can provide information about both size and shape, and because of the relatively small size of Fasciola intermediate forms (Fig. 2), it was expected that this sample could be identified using traditional techniques. Nevertheless, linear measurements did not just reflect global size variation since they could also allow an excellent score of recognition for the pair F. hepatica and F. gigantica, which are parasites of similar global size (Fig. 2, Table 4). The extraction of shape (LSR) did not produce different results, confirming the relatively poor efficiency of shape removal from our linear measurements. With or without shape extraction, traditional morphometrics appeared to be a powerful means of species determination. It does, however, require an excellent knowledge of the parasite’s internal anatomy.

The previously recommended number of linear measurements necessary to distinguish F. hepatica from F. gigantica and Fasciola intermediate forms was as high as 20 (Valero, Panova & Mas-Coma, 2005; Ashrafi et al., 2006). By considering the contribution of each variable to their first two PC, we were able to reduce this number from 20 to 12, with similar or better results (see Table 3). This improvement could be explained by the elimination of redundancy among measurements, and a possible lack of precision for some of them. Either way, our examination of repeatability produced satisfactory results for both shape (93%) and size (99%), although these results were lower than the results of the outline-based technique (98% and 99%, respectively).

The outline-based morphometrics analysis was based on manual contour digitization. Although this type of manual data collection is relatively fast, an automated capture technique might be preferable. Automated curve tracing is already included in many types of imaging software, and it may be more appropriate for generalizability (Yang et al., 2015), but its useful application is dependent on the complexity of the contours (Sheets et al., 2006; Firmat et al., 2010). We would recommend that a study be performed that is specifically aimed at comparing automatic and manual capture techniques. The fact remains, however, that whether manual or automatic techniques are used, the outline digitization process is a relatively easy task and does not require highly specialized skills from the user.

The landmark-based approach required a little more knowledge about the internal anatomy of the parasites and did not provide satisfactory results. The lower reclassification scores of the landmark data relative to the OD should not be regarded as a rule. In our case, very few reliable LM could be suggested, reducing the amplitude of shape capture, which could explain the unsatisfactory scores obtained. It is possible that better results might be obtained with, for instance, a different landmark configuration, or a different number of landmarks.

In our validated method of reclassification each individual was sequentially removed from the total sample and its identification performed using the model computed without it, a method also called a “leave-one-out” or “jack-knife” procedure (Manly, 1986). Thus, individual identification was repeated as many times as there were individuals in the total sample (see Data S1). Our “accuracy” for each method (Tables 3 and 7) was the percentage of individuals that were correctly identified at the end of each such procedure. For the ANN classification, each classification session was repeated 30 times, and the accuracies were presented as the average and standard deviation of the 30 sessions (Tables 3 and 7).

The validated classification applied to different kinds of data used different methods (Table 3). The one based on the Mahalanobis distance (Dist) generally produced higher scores with more correct identifications than the other classification techniques, which conforms with previous studies and highlights the frequent relative superiority of this technique (Dujardin, Dujardin & Kaba, 2017; MacLoad et al., 2007; Sonnenschein et al., 2015). However, reliable classification and assessment of group separation using this analysis is based on several assumptions, some of which frequently do not coincide with morphometric data (Mitteroecker & Bookstein, 2011). For this reason, we added additional classification techniques, such as the simple maximum likelihood (MxL) approach (Polly & Head, 2004; Dujardin, Dujardin & Kaba, 2017) and the ANN technique.

Since the MxL method assumes normally distributed data and independence among the characters used, our best results were obtained when applied to (a subset of) the PC of the log-transformed input variables, rather than when applied to the raw variables (detailed comparisons not shown). In agreement with previous comparisons (Dujardin, Dujardin & Kaba, 2017), the MxL technique provided a slightly lower accuracy than the Dist method.

In our study, the ANN produced slightly lower scores than the Dist or the MxL approaches. Our idea was to compare the taxonomic signal embedded in different types of data rather than to deeply explore the power of each classification method. So, for each classification method, we tried to subject each type of data to the same process. The neural network was thus configured in the same way, irrespective of which data were used as the input (see Data S2). Certainly, some improvements could be obtained by refining the configuration according to the kind of data being used. Moreover, a possible explanation for the relative weakness of the ANN method used here is the low sample sizes of the training data. The ANN method may require many data to achieve an accurate level of performance (Soda, Slice & Naylor, 2017). However, the general scores were satisfactory, and we showed that this technique could greatly benefit from using combined data as the input, for example using a few traditional measurements together with outline shape data (Table 7).

Conclusion

Both traditional and modern morphometric approaches can be helpful in the morphological identification of Fasciola sp. We showed that by reducing the number of linear measurements from the 20 commonly recommended ones to a selected subset of 12 measurements, the accuracy of identification could be improved.

In our study, the outline-based approach produced satisfactory classification scores, more so than the landmark-based approach. Moreover, we showed for the first time the apparent differences in shape between “intermediate forms” of Fasciola sp. and F. gigantica and F. hepatica. The relative simplicity of data collection (even manual collection) makes the outline-based approach an attractive alternative.

By comparing various data collection and classification techniques, our study suggests promising new strategies to follow in helping with the identification of species in the genus Fasciola based on their morphology.

Supplemental Information

Figure S1 The comparison morphological characters measurement of this study and other published data

Data were compared graphically with previously published ones, showing the departures of observed means (published ones and ours) from the total means

Click here for additional data file.

Figure S2 PCR-RFLP patterns of ITS1 amplicon (680 bp) digested by Rsa I enzyme

Lane 1: 100 bp DNA ladder, lane 2 to 3: F. gigantica, lane 4: Fasciola intermediate form, lane 5 to 14: F. gigantica, lane 15: Fasciola intermediate form. The PCR digested amplicons were run on 2% agarose gel at 50 V for 180 min.

Click here for additional data file.

Figure S3 PCR-RFLP patterns of ITS2 amplicon (550 bp) digested by Nla III enzyme

Lane 1: 100 bp DNA ladder, lane 2 to 3: F. gigantica, lane 4: Fasciola intermediate form, lane 5 to 14: F. gigantica, lane 15: Fasciola intermediate form. The PCR digested amplicons were run on 2% agarose gel at 50 V for 180 min.

Click here for additional data file.

File S1 Raw data of morphological measurement of Fasciola spp

Click here for additional data file.

Data S1 Classification algorithm

Click here for additional data file.

Supplemental Information 6 Data S2

Click here for additional data file.

The authors would like to thank the Center Equipment Unit, Faculty of Tropical Medicine, Mahidol University for supporting the excellent imaging system. We would like to thank Mr. Ulf Biallas, Software Engineer, Axel Springer Ideas Engineering GmbH for his kind assistance.

Additional Information and Declarations

Competing Interests

Author Contributions

Data Availability

The authors declare there are no competing interests.

Suchada Sumruayphol, Praphaiphat Siribat, Jean-Pierre Dujardin and Urusa Thaenkham conceived and designed the experiments, performed the experiments, analyzed the data, prepared figures and/or tables, authored or reviewed drafts of the paper, and approved the final draft.

Sébastien Dujardin and Chalit Komalamisra conceived and designed the experiments, analyzed the data, prepared figures and/or tables, authored or reviewed drafts of the paper, and approved the final draft.

The following information was supplied regarding data availability:

The raw measurements are available in the Supplemental Files.

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
