# Peer review of "Fasciola gigantica, F. hepatica and Fasciola intermediate forms: geometric morphometrics and an artificial neural network to help morphological identification"

_PeerJ, doi:10.7717/peerj.8597_

## Round 0.1 · original submission · Major Revisions

Both reviewers have raised several comments that require careful attention in a revision. Many of these comments indicate that some details, especially in the methods and relating to how, precisely, the neural network steps were implemented, are absent in the current version of the text. These omissions negatively impact the ability of the reader to understand the methods and the validity of their application in this setting. I encourage the authors to address the questions raised by reviewer #1 in relation to improving the methods section of the paper. I also agree that terminology needs to be revised throughout the manuscript to reflect current usage in the geometric morphometric literature.

·

Basic reporting

There are numerous grammatical errors throughout the text that require thorough revision, preferably by a native English speaker. While these errors do not impede the understanding of the paper’s content, they greatly affect the professional presentation of the study and require further revision.

The introduction provides a well referenced contextualisation of this study, especially from a biological standpoint, yet regarding the field of geometric morphometrics, a number of terminological components should be revised. These include (in no particular order of importance):

Between lines 97 and 103 the authors mention the geometric and mathematical problems presented by the variable size when considering traditional metric approaches in biological studies, yet do not mention the terminological differentiation between methods that exclude size from those that include this variable. I recommended that the authors clarify the difference between shape (which is correctly mentioned) and form (which is not mentioned at any point within the text) to help construct a stronger argument within this paragraph. For this I recommend the authors consult the work of Jungers, Falsetti and Wall (1995) with their paper titled Shape, Relative Size, and Size-Adjustments in Morphometrics in the Yearbook of Physical Anthropology, volume 38, pages 137-161.

A similar terminological issue can be found in line 110, where the authors mention Rohlf and Marcus’s methods for “visually representing shape”. It is important to point out that this technique has a specific name that should be mentioned and was in fact developed prior to Rohlf and Marcus’ work. I recommend that the authors consult the ample bibliography regarding Deformation Grids and Thin Plate Splines while also encouraging the authors to use these terms within their own work.

The authors frequently make reference to “outline-based” approaches. Once again I encourage that the authors use the correct terminology here. Throughout the methods, results and other parts of the text, the authors make reference to the “Outline Data” and “Pseudolandmarks”; the correct term the authors should be using is semilandmarks. For the differentiation between landmark types I recommend the authors consult the works of Dryden and Mardia (1988) in their book titled Statistical Shape Analysis. As for the terminological definition and developed use of semilandmarks the authors are encouraged to consult the work of Dr. Philip Gunz in collaboration with P. Mitteroecker and F.L. Bookstein, especially their 2004 book chapter titled Semilandmarks in Three Dimensions from D.E. Slice’s (Ed.) Modern Morphometrics in Physical Anthropology (pages 73-98). As far as I am aware, Gunz et al.’s book chapter is available online via Research Gate.

On line 257 the authors make reference to: “final outputs were the “weights””. This use of the term “weights” is quite confusing especially when referred to as a product of Neural Networks. Machine and Deep Learning models use weights to obtain the outputs. The whole process of training a model is to find the optimum combination of weights throughout the model that best map out the relationships between inputs x and outputs y (at least in the case of supervised learning). The outputs in themselves are not the weights of the model. The authors are therefore required to clarify what they refer to by the term weights? I suggest the authors either use a different term for the outputs or revise this sentence entirely to clarify what they actually mean. The presentation of this particular statement currently generates quite a lot of confusion.

In line 261 it is not entirely apparent as to what the authors refer to by using the terms “testing” and “training”. Good practice in Machine and Deep Learning require that testing and training sets be separated prior to training. This is performed so as to ensure the model is unbiased when presented with new data, thus providing an efficient means of evaluating model performance. Within this paragraph the authors talk about random subdivisions throughout each iterative process, therefore I think the authors are actually referring to training and validation sets. In different types of cross-validation techniques this is the correct term that should be used. I recommend that the authors revise this statement to ensure they are not confusing the term testing with validation.

While the figures and tables presented are of great value and help contribute towards the detailed description of the results, the authors are encouraged to revise the titles and descriptions provided for each. In table 2 the authors use superscript letters to indicate statistical differences according to p values, however at no point mention what a, b, c or a combination of these letters actually means. Is each letter an indicator of the degree of statistical significance? If so is "a" more significant than "c"? Therefore what does the combination of "a" and "b" mean? In table 3 a slightly better description of these superscript letters is provided, however it is still unclear as to what the authors are referring to. Therefore (1) would it not be better to directly include the p values (highly recommended) and (2) why is this description provided for table 3 and not for table 2? I suggest the authors review each of their table and figure descriptions in detail to improve the presentation of their work.

On a positive note, I think the supplemental materials and raw data supplied are of value to the scientific community and may help authors in the future when performing similar studies.

As far as I can tell, the reference list is well structured, complete and all citations mentioned within the reference list are indeed included within the text.

Experimental design

The use of Artificial Neural Networks within this study requires more details, not only in able to support the validity of results obtained but also to ensure that other authors are able to replicate these results.

The authors mention they used a single hidden layer composed of three neurons after a process of trial-and-error. In order to properly evaluate the performance of artificial networks in the processing of data of this nature much more information is needed regarding the configuration of the model. How was the model trained? Which optimisation formula was used? Were any further hyperparameters tested and if so which parameters were changed? Considering this, the authors are encouraged to provide much more data regarding the training of their neural network, which could easily be provided if necessary as an additional supplementary file.

In addition to this, the authors mention firstly that the neural network was only able to achieve 64% in one of the cases, while tables also demonstrate how the neural network performed poorly in other cases. This to me does not seem to be an issue with the use of neural networks in these type of studies, but more to do with the configuration of the network and/or the quality of input data.

While Deep Learning was not the primary objective of this paper it is useful to comment that a more detailed study into the architecture of the model, the correct configuration and hyperparameter optimisation may be the best means of significantly improving the results. Along these lines I encourage that the readers consider this concept further and comment within the discussion of their paper the possibilities this experimental factor may have on their result. Please consult, for example the works of Wolpert (1996) The Existence of a Priori Distinctions between Learning Algorithms.

Why were only 8 PC scores used? How many PC scores in total were produced, and why were only these selected for classification? Did the authors perform any study into the value and weight each PC score had on the quality of the results and efficiency of the models?

Turning to the Geometric Morphometrics a number of important issues have to be questioned:

As before I strongly suggest the authors justify their use of only 8 PC scores.

How many semilandmarks were used for the study of “Outline Data”? Why this number? Where other configurations tried and tested? Why were they not combined with the fixed landmark data? Consulting traditional geometric morphometric literature, most authors do not perform their analyses solely using fixed landmarks or solely using semilandmarks, yet choose to combine the different types of configurations so as to maximise resolution and accuracy. If the authors used a model that contained both the fixed landmarks and the semilandmarks describing the contour, they may have a better means of defining morphological variations within their samples.

Validity of the findings

The results within this study appear conclusive, however a number of important concepts have to be confronted in order to improve the presentation of the authors’ results.

The act of basing one's arguments solely on a given p value is not necessarily good practice within applied statistics and has been the topic of heated debates for quite some time. The authors are thus encouraged to support their results with additional statistical metrics that can support their claims. For example,

In ANOVA testing and other univariate or multivariate tests should be accompanied either by data regarding residuals or the F statistic, all of which are useful for supporting results rather than relying solely on a given p value.

Mahalanobis distances – The authors mention briefly the significance of mahalanobis distances obtained, yet make no reference as to the precise value of these distances themselves. This can neatly be summarised in a table containing the mahalanobis distance between each group and their corresponding p value of statistical significance.

The authors when referring to the classification results only make reference to the accuracy. This is a poor means of summarising the performance of a model. First of all, is this a balanced accuracy from the validation technique employed, or is this the accuracy observed when tested on the test set? How was this value obtained? Neural Networks are stochastic in nature, different training sessions are likely to produce different results: have the authors taken this into consideration? Examples of other model evaluation metrics that should be considered are specificity and sensitivity values obtained through the calculation of confusion matrices. Furthermore, what was the overall confidence of the classifications? It is not the same if a model classifies an individual correctly with 51% confidence as opposed to a model that correctly classifies an individual with 99% confidence. This can be consulted via each model/test’s loss when making overall predictions.

Why did the authors choose to use Mahalanobis distance calculations as opposed to Procrustes distance calculations? Mahalanobis distances are well known for being affected by sample size. This is especially important considering the issue of sample size is one that is mentioned by the authors themselves on a number of occasions.

Along these lines, the authors mention that 36 complete specimens were selected for the study (lines 187-188), yet previously mention that around 90 worms were collected in one of the samples (line 125). Why were 36 only used at this later stage? Could the authors please simply clarify where these numbers have come from?

In addition, why choose 36? Is this number statistically significant? How did the authors ensure this? Were statistical power tests performed to confirm 36 as a statistically significant value?

Regarding the conclusions, in lines 421-422 the authors state that “landmark data did not allow us to recommend this morphometric technique for identification purpose”. I have to disagree with this statement and ask that the authors revise. In this particular study the landmark model was not as successful but this is not to say that if another landmark configuration is used other authors may not conclude differently. The quality of these results may vary depending on how the data is processed (consider previous comments regarding the ANN) and the combination of landmark models used (as previously stated in other comments). The authors should refrain from rejecting landmark based models considering how another configuration may more useful when combining both semi and fixed landmark data.

In lines 389 the authors mention that “the landmark based approach require a little bit more knowledge about the internal anatomy of the parasites…”. Have the authors considered possible statistical noise product of inter-analyst errors? The value of some landmarks can be assessed on the reproducibility of the landmark model.

Additional comments

The study presented by Sumruayphol, Siribat and colleagues demonstrates how multiple statistical methods can be used to identify different, with the use of both traditional metric techniques, geometric morphometric techniques as well as numerous statistical and artificially intelligent means of processing these different datasets. The ideas presented within this paper are interesting while their results are also valuable, nevertheless, there are elements within the methods and results that require further development in order to provide a more complete and precise presentation of their results. Likewise some methodological questions need to be clarified to justify their use of the specific techniques employed within this paper. This could greatly contribute towards the value of their study. Under this premise I recommend this paper undergo major revisions before it be considered for publication.

Reviewer 2 ·

Basic reporting

Clear English used throughout, literature references properly cited, raw data shared.

Experimental design

Methods described with sufficient detail and information to replicate.

Validity of the findings

All underlying data have been provided; they are robust and statistically sound.

Additional comments

This manuscript is informative about characterization and identification of Fasciola gigantica, F. hepatica and Fasciola intermediate forms. It is an interesting study that would be useful for improve the identification and, in my opinion, scientifically sounds. The authors present their data and associated analyses in a clear and coherent manner. I believe that the conclusions drawn are well-supported by the data included.

Below authors will find some suggestions/questions:

Is the ethics committee's certificate available regarding the use of animals in this study?

Line 54 – typo “Mas-Coma, Bargues &Valero, 2005”. Please check other typos throughout the manuscript.

Line 69 – “...has been used to help Fasciola species identification” insert some references here.

Line 80 – I don't think "Amer" should be italicized here.

Line 81 – When you cite the acronym for the first time please explain what it means (just in the first time).

Line 88 – The acronym ITS1 and ITS2 has already been cited above and should be explained just for the first time in the text. Here one should only refer to the acronym.

Line 86 – This paragraph is similar to the previous one, with many repeated information. It would be better to restructure both.

Line 94 – “(Periago et al., 2006; Valero, Marcos & Mas-Coma, 1996; Valero, Panova & Mas-Coma, 2005; Ashrafi et al., 2006; Afshan et al., 2014)” Should these references be in alphabetical or chronological order? Because neither is being followed. Check this for the other text references.

The authors should make explicit their hypotheses at the end of the Introduction section.

Materials and Methods:

Line 141: Regarding specimens obtained from the Liverpool School of Tropical Medicine (England): Do the authors know how this material was prepared? Could a procedure increase or decrease its size or change its shape in relation to specimens collected in the field? The time (1993) could not change the specimens?

Line 187: “We selected 36 complete specimens”, but you collected 90 specimens, right? Make it clear why you didn't use them all.

Line 202: check the citation of Valero et al and the following references.

Line 255: I think the correct citation of “Camila, Antonio & Lincoin, 2015” actually is “Lorenz, Ferraudo & Suesdek, 2015”.

Line 280: Is this molecular identification method 100% accurate? Or is there any overlap between the identifications?

Line 305: “Fasciola intermediate forms” letter S is blue.

---

## Round 0.2 · Minor Revisions

Thank you for addressing the previous set of comments to your manuscript. One of the reviewers, who raised some substantial issues on the previous version, has re-reviewed your manuscript and has commented positively that the revision is much improved. They have made some further minor suggestions that need to be addressed. With regard to the PC scores, I suggest the authors consider the broken stick model (Jackson, 1993) regarding choosing components that represent significant proportions of variance in the the sample. Please see:
Jackson, D. A. (1993). Stopping rules in principal components analysis: a comparison of heuristical and statistical approaches. Ecology, 74, 2204-2214.

·

Basic reporting

The overall presentation of the paper has improved significantly. The presentation of both tables and figures has also improved, however figures 2 needs a label for the axis represented. I would also suggest including the units of measurement in said label to facilitate the reading of the graph. While it is understandable what figures 3 and 4 represent, I would also suggest labelling these axes and including the percentage of variance represented in each PC score.

Experimental design

While the inclusion of supplementary files and expansion of the methods is an important improvement to the quality of the paper, there are still a number of issues that have to be addressed to ensure reproducibility. In my previous review I recommended that the authors describe the hyperparameters used in their artificial neural network. While many of my doubts have initially been clarified, the authors are still missing some details. It is important that the authors at least describe the optimization algorithm used. The authors are also required to state which activation function is used in the single hidden layer as well as the activation function used to produce the output. The importance of these three details (at least) is found in how any analyst who wishes to replicate this study using a different software would need to know how to configure and train the classification model. This is a vital piece of information that any user of python or R (such as myself) would require in order to construct the neural network and replicate these methods.

Validity of the findings

In my previous review, I asked the authors to justify their reasons for selecting their number of PC scores. This comment simply required the authors state the reason behind their use of this specific number of PC scores. The authors’ choice of wording however raises more questions than it does answers, with their stress on the use of the word ‘ “rule” ‘. Why is “rule” in quotation marks? If the authors wish to state that they followed a ‘ “rule” ‘ in statistical multivariate analysis, could they please cite an author who states this ‘ “rule” ‘ in order to justify their choice in phrasing. Multivariate statistics is a very large field of research and no single ‘ “rule” ‘ exists. Otherwise I think that simply removing this part of the sentence would help avoid issues. They could simply state “This specific number of PC scores was selected so as to use a number of variables that does not exceed the number of individuals in the smallest sample” or a phrase along these lines.
A small note: while the inclusion of the mean and standard deviation of 30 iterations of the Neural Network are a valuable inclusion, could the author please specify whether the sd value is the first deviation from the mean (68% of the distribution) or otherwise (2sd = 95% distribution, 3sd = +99% distribution).

Additional comments

The quality of the paper has improved significantly and I thank the authors to take the time to explain each of the questions/revision comments raised. These minor revisions are relatively easy to rectify and each can be corrected with the addition of a few words that clarify the respected sections of the paper. Upon clarification of these minor points I feel this paper would be ready for publication.

---

## Round 0.3 · accepted · Accept

Thank you for addressing the minor comments raised by the reviewer in the last round of revision. Your responses have fully addressed the reviewer's comments and your paper is now ready to be accepted. I note one typo in your added text - please check manuscript line 510 and correct to, "This specific..".